# Validation and Evaluation of a Behavioral Circuit Model of an Enhanced Electrostatic MEMS Converter

**DOI:** 10.3390/mi13060868

**Published:** 2022-05-31

**Authors:** Mona S. Salem, Abdelhalim Zekry, Mohamed Abouelatta, Ahmed Shaker, Marwa S. Salem

**Affiliations:** 1Electronics and Communications Engineering Department, Faculty of Engineering, Ain Shams University (ASU), Cairo 11517, Egypt; monabasyoni@gmail.com (M.S.S.); aaazekry@hotmail.com (A.Z.); m.abouelatta@eng.asu.edu.eg (M.A.); 2Physics and Mathematics Engineering Department, Faculty of Engineering, Ain Shams University (ASU), Cairo 11517, Egypt; 3Department of Computer Engineering, Computer Science and Engineering College, University of Ha’il, Ha’il 55211, Saudi Arabia; marwa_asu@yahoo.com; 4Department of Electrical Communication and Electronics Systems Engineering, Faculty of Engineering, Modern Science and Arts (MSA) University, Cairo 12566, Egypt

**Keywords:** behavioral, circuit model, MEMS converter, vibration, energy harvesting, *C*–*x* curve

## Abstract

In this current study, the validation and evaluation of a behavioral circuit model of electrostatic MEMS converters are presented. The main objective of such a model is to accurately find the converter behavior through the proper choice of its circuit elements. In this regard, the model enables the implementation of the electrostatic MEMS converter using commercially available off-shelf circuit elements. Thus, the overall vibration energy harvesting system can be implemented and tested without the need for fabricating the converter. As a result, the converter performance can be verified and evaluated before its fabrication which saves the expenses of fabricating trailed prototypes. To test the model, we apply it to an enhanced converter in which the conventional electrostatic MEMS converter is modified by depositing the tantalum pentoxide, Ta_2_O_5_, a high dielectric constant material, on its fingers’ sidewalls. Such a deposition technique causes an appreciable increase in the overall converter capacitance and, in turn, the output power, which is boosted from the range of µw to the range of mW. Next, the converter behavioral circuit model, which is based on representing its capacitances variations with respect to the input displacement, *x* caused by the vibration signal, *C*–*x* curve, is built up. The model is qualitatively validated and quantitatively evaluated. The enhanced converter performance is investigated through the interaction of its model with the power conditioning circuit. From the simulation results, it is revealed that the converter behavioral circuit model accurately accomplishes the vibration energy conversion operation. As a result, the specification of the required controlling pulses for the converter operation is accurately determined. Finally, the model accuracy is validated by calibrating its performance with a traditionally simulated and fabricated electrostatic MEMS converter.

## 1. Introduction

During the previous two decades, a significant enhancement in the development of low-power, portable, small size, and remote devices has been achieved. This enhancement causes the replacement of the traditional with nontraditional power sources. Recent research concentrates on harvesting energy from the environment. The reason is that, in some applications which are in an inaccessible environment or which have high maintenance costs, energy harvesting becomes essential. Wireless sensor nodes in remote areas, biomedical devices, and implanted health trackers are examples of applications that require energy harvesting systems [1,2,3,4]. Energy harvesting is the direct conversion of environmental energy such as thermal, wind, vibration, and solar energy into electrical energy [5,6,7,8]. A MEMS vibration energy harvester is mainly a spring-mass system. It has one or more resonance frequencies of vibration. Based on the mechanism of transfer, most MEMS vibration energy harvesters are classified into three different types: piezoelectric [9,10], electromagnetic [11,12], and electrostatic harvesters [13]. In an electromagnetic energy harvester, the losses caused by its coil resistance are one of its major drawbacks. In addition, it requires complex fabrication processes which have low process compatibility [14]. Electrostatic harvesters, on the other hand, can be easily fabricated using standard micro-machining processes [15,16,17,18].

It is important to determine which MEMS transducer type satisfies the best power density in a certain energy harvesting system. It is shown that, at low accelerations, electrostatic harvesters are better than piezoelectric harvesters because of the lower energy losses and vice versa at high accelerations. At very high accelerations, the performance of piezoelectric energy harvesters is severely degraded because of the dielectric breakdown limit. Thus, the electrostatic harvesters are better at very high accelerations. Based on such a comparison, it is clear that the optimal transduction mechanism depends on the harvesting acceleration, operating frequency, and device size [19,20,21,22,23,24,25]. Moreover, electrostatic harvesters are based on the capacitive sensing mechanism, which is the main dominant method for micromachined applications. The reason is that it is compatible with all the fabrication approaches and stiffness [26,27,28,29]. Recent research studies focus on enhancing the electrostatic MEMS converter performance by mainly increasing its output power. Unfortunately, the converter output power is still in the range of µW to fractions of mW [30,31,32,33]. Moreover, there is another direction in recent research that focuses on the accompanying circuit, which is necessary for delivering the output power of the converter to the load, the power conditioning, and the power controlling circuits. Also, such a direction still needs more research efforts [34,35,36,37].

Our research group conducted previous efforts in this field. We had already developed a proposed model for the electrostatic MEMS converter, which satisfied the charge transfer between the converter model and the system storage element. However, it did not achieve a good agreement between the calculated converter output power and the simulated output power gained from the vibration energy conversion operation [38,39,40]. Furthermore, the transitions between the system elements need to be restudied and re-simulated as they were not exactly representing the specific operation during each transition. In addition, in our previous work, the output power that was achieved from the designed converter was relatively low, 0.122 mW [38,39,40].

The main contribution of this research work is to reinvestigate our promising proposed converter behavioral circuit model. Thus, by using the enhanced proposed model, the difficulties of practically testing the designed converter due to the unavailability of the fabrication facilities could be overcome. Furthermore, the electrostatic MEMS converter requires deep reactive ion etching (DRIE) for large aspect ratios [41,42], which is considered a difficult fabrication process to be available in most cleanrooms. Thus, in this work, firstly, the performance of the conventional electrostatic MEMS converter is improved by depositing the high dielectric material, Ta_2_O_5_, on its sidewall fingers. As a result, the converter output power is boosted to the range of a few mW which is considered a remarkable enhancement when compared with the most recent research work [30,31,32,33]. Secondly, the main contribution of this work is to illustrate the converter behavior based on its *C*–*x* curve, which demonstrates the converter capacitance changes with respect to the displacement, *x*, caused by the input vibration signal. Thus, the performance of our previously proposed behavioral circuit model for the MEMS converter can be enhanced. Such enhancement aims to achieve a good agreement between the calculated converter output power and the simulated output power gained from the vibration energy conversion operation. As a result, our proposed model will be adaptive to be applied to any other MEMS converter. Thus, by using this model, the implementation and testing of the MEMS converter along with the overall vibration energy harvesting system using commercially available off-shelf components can be easily achieved. As a result, the designed converter performance can be cheaply tested and evaluated before going through the costly fabrication procedures. Before implementing the model, it must be qualitatively validated and quantitatively evaluated to confirm its vitality in well demonstrating the converter performance.

Moreover, the interaction between the model and the power conditioning circuit enables the accurate specification of the required controlling pulse for the converter operation. As a result, the converter conditioning and controlling circuits, along with the whole vibration energy harvesting system, can be easily designed, implemented, and tested without the need for sophisticated analytical modeling [34,35,36,37].

The paper is organized as follows. In Section 2, the electrostatic MEMS converter, along with its main technological parameters, are presented. In Section 3, the main governing analytical equations for the converter performance are illustrated. Moreover, the maximum voltage, *V_max_*, which is one of the main system constraints, is simulated and determined. After that, in Section 4, the electrostatic MEMS converter behavior based on its *C*–*x* curve is qualitatively explained. Then, in Section 5, the proposed behavioral circuit model of the converter is given. Section 5 illustrates the qualitative validation and the quantitative evaluation of the converter model. Finally, a summary of the important findings and conclusions of this work is offered in Section 6.

## 2. Electrostatic MEMS Converter and Main Parameters

The in-plane gap-closing electrostatic MEMS converter is considered the most commonly used type of electrostatic MEMS converter as it gives the highest power density [25]. Figure 1a demonstrates the conventional in-plane gap-closing electrostatic MEMS converter, while Figure 1b shows the details of its main technological parameters, respectively [42]. The converter comb drive technological parameters are identified in Table 1. Such parameters will be used in building up the converter behavioral model in the coming sections. The converter comb drive has two main capacitances according to its fingers position with respect to its fixed comb and anchors, which are minimum capacitance (*C_min_*) and maximum capacitance (*C_max_*), expressed by Equations (1) and (2), respectively.
(1)$$C_{min}=\frac{4N_{g}\varepsilon_{0}\varepsilon_{r}L_{f}t}{d_{nom.}}$$
(2)$$C_{max}=\frac{4N_{g}\varepsilon_{0}\varepsilon_{r}L_{f}td_{nom.}}{\left(d_{nom.}^{2}-Z_{max}^{2}\right)}$$
where: *N_g_* is the number of fixed/movable fingers; *d_nom._* is the nominal gap between fingers at rest position, and *Z_max_* is the maximum deflection of the converter springs from their rest position [42]. In this work, the technological parameters of the converter shown in Figure 1b are selected as a case study to investigate its performance [19,43].

Referring to Table 1, the main objective of applying such a case study is to increase the converter capacitance; thus, its output power will increase. Based on the case study [19], to increase the converter capacitance, the nominal gap between its fingers at the rest position (*d_nom._*) should be minimized. Its minimum value is limited to 7 µm based on the fabrication constraints [19]. Moreover, the converter thickness (*t*) should be increased. It is set to be 500 µm as a case study [19] which satisfy the required enhancement of the converter capacitances, taking into consideration that it requires a lower fabrication cost. Moreover, the finger length (*L_f_*) should be maximized; therefore, it is set to be 512 µm considering that it is limited by the spring deflection. Concerning the finger width (*W_f_*), it is selected to be 7 µm which ensures proper operation of the converter [19].

Further, the shuttle mass length (*L_m_*) should be increased; however, it is hindered by the spring and the converter dimensions. The optimal values for *L_m_* and *W_m_* are found to be 1 cm by 0.3 cm, respectively [19]. The dielectric distance (*d*) is assumed to be 50 µm to reduce the stray capacitance [43].

## 3. Main Governing Equations of the Electrostatic MEMS Converter

In this section, the main analytical equations which govern the converter performance are illustrated. Based on Figure 1, during the input vibration, the converter movable combs and spring move in the lateral direction. The capacitance changes due to the change of the dielectric gap between the combs drive fingers. The converter output power is given by Equation (3) [42,44].
(3)$$P_{out}=2f_{0}E$$
where *P_out_* is the output power of the converter, *E_useful_* is the useful energy per cycle and *f*_0_ is the driving frequency of the input vibration source, which is also the converter resonant frequency. The converter is designed to operate based on the source of the input vibration taken from a gas turbine. Thus, the converter can be used in several different modes in critical industries such as power generation, oil and gas, process plants, aviation, as well domestic and smaller related applications [45,46].

In Equation (3), the factor of 2 is added to represent both the converter charging and discharging operations for each vibrational cycle [42,44]. The useful energy per cycle from the converter is given by Equation (4),
(4)$$E_{useful}=\frac{1}{2}(C_{min}V_{max}^{2}-C_{max}V_{min}^{2})$$
where *V_max_* is the maximum voltage, which is one of the basic key factors of the vibration energy harvesting system [19,33], it is determined by the converter and its power conditioning circuits. It has two limiting factors, the breakdown voltage of the power conditioning circuit, MOSFET switches, and the breakdown voltage of the converter. In this work, the 0.35 µm CMOS technology is used for designing the power conditioning circuit switches. From such a technology file, the breakdown voltage for the power MOS transistors is equal to 18 V. Thus, *V_max_* must be less than 18 V. For the design safety, *V_max_* is assumed to be half of the power switches breakdown voltage, i.e., 9 V [47].

Furthermore, *V_max_* must be less than the converter breakdown voltage, which is determined by the breakdown of the air existing between the converter fingers. Thus, the converter’s maximum electric field caused by *V_max_* has to be less than the breakdown electric field of the air, which is 3 × 10^6^ V/m [48].

To be able to determine the optimum value of *V_max_*, two fingers of the converter are simulated using the MATLAB PDE toolbox. The maximum electric field, *E_max_*, between two fingers of the converter is simulated at different input voltage (*V_ip_*), 2, 4, 6, 8, 10 V. The optimum *V_max_* is determined based on the conditions discussed herein. Figure 2 represents the electric field distribution and the maximum electric field (*E_max_*) between two fingers of the converter for different input voltages. It is obvious that at *V_ip_* equals 10 V, *E_max_* is found to be 2 × 10^6^ V/m, which is considered a risky value as it is close to the air breakdown electric field. Thus, to achieve a safer design, the optimum *V_max_* must be set to 8 V, at which the maximum electric field (*E_max_*) between the converter fingers is 1.6 × 10^6^ V/m, which guarantees a safe design. Such a design value of the optimum *V_max_* also satisfies the safe design concerning the breakdown voltage of the converter power conditioning circuit power switches [47]. Figure 3 represents *E_max_* between the converter fingers at different input voltages. The figure highlights that at the optimum *V_max_*, which is 8 V, *E_max_* is 1.6 × 10^6^ V/m which promotes the converter design.

Referring to Equation (4), *V_min_* is the converter’s initial voltage that exists on the converter at the beginning of each input vibration cycle to start the conversion operation. A reservoir capacitor is used to satisfy such an important requirement, as will be explained in the coming sections. In this work, the converter operation was based on the charge constrained technique which has the advantage of using one single initial voltage source to start the operation and satisfying that *V_min_* << *V_max_* [33].

Thus, only one single charge source is needed to begin the process, and its value was less than *V_max_* [33]. Based on the charge constrained conversion technique, *Q* was the charge placed on the converter fingers while its capacitance was at a maximum value. When the fingers are separated apart, the capacitance decreases until it reaches *C_min_* while the voltage increases to *V_max_*. *Q* is held constant according to the charge constrained conversion technique [33], where it is expressed by Equation (5),
(5)$$Q=C_{max}V_{min}=C_{min}V_{max}$$

By substituting Equation (5) into Equation (4), *E_useful_* is expressed by Equation (6)
(6)$$E_{useful}=\frac{1}{2}\left(\frac{Q^{2}}{C_{min}}-\frac{Q^{2}}{C_{max}}\right)$$

By combining Equations (3) and (6), the final expression for the converter output power is given by Equation (7):(7)$$P_{out}=f_{0}\left(\frac{Q^{2}}{C_{min}}\right)\left(1-\alpha \right)$$
where *α* = *C_min_*/*C_max_*. Based on Equation (7), the converter output power could be increased by increasing its *C_max_*; thus, the factor *α* will be decreased. Further, the converter output power could be increased by increasing the charge *Q*.

Next, by substituting *Q* from Equation (5) in Equation (7), the converter output power final expressed was given by Equation (8)
(8)$$P_{out}=C_{min}V_{max}^{2}(1-\alpha )f_{0}$$

To calculate the output power (*P_out_*) of the converter according to the case study used in this work, recalling Table 1, *C_min_* and *C_max_* were calculated to be 8.8 nF and 0.6 nF, respectively. The value of *α* was calculated to be 0.07. Substituting *V_max_*, *f*_0_, and *α* in (8), the maximum calculated *P_out_* for the used case study is found to be 0.09 mW.

Now, in order to improve the converter output power, we propose the deposition of a thin layer of a high dielectric constant material, tantalum pentoxide (Ta_2_O_5_), on the converter comb drive fixed and movable fingers. Such deposition increases the converter’s overall capacitances resulting in increasing the converter output power. Moreover, the deposited Ta_2_O_5_ acts as electrical stoppers which overcome the short circuit condition of the converter when its fingers come together. Figure 4 shows a simplified demonstration of the converter fixed and movable fingers with the deposition of the Ta_2_O_5_ thin layer.

The Ta_2_O_5_ material was selected for this case study as it has remarkable electrical and dielectric properties. It acts as an electrical insulator that has a high dielectric constant in the range of 25 to 35 [49]. The value of the Ta_2_O_5_ dielectric constant was set to the minimum value of 25 as a worst-case design for the required deposited thickness (*d_min_*), which is 0.25 µm. This thickness was large enough to avoid the leakage problem that may occur when the dielectric materials layer thickness is in the range of nm [49,50,51]. The converter output power, Pout, was recalculated based on the effect of depositing the Ta_2_O_5_. It became 2.3 mW. Figure 5 shows the converter output power in the two different cases, with and without the deposition of Ta_2_O_5_ on the converter fingers sidewalls. It is clear that the output power increased in the case of deposition. In the coming sections, the values of the enhanced converter capacitances with depositing Ta_2_O, will be used in the validation and evaluation of the behavioral circuit model.

## 4. Proposed Behavioral Circuit Model of the Electrostatic MEMS Converter

In this section, our proposed behavioral circuit model [38,39,40] of the electrostatic MEMS converter is reinvestigated based on its capacitance variations with respect to the input displacement, *x*, caused by the vibration signal. Firstly, the converter behavior based on its *C*–*x* curve is qualitatively illustrated. Secondly, the converter proposed behavioral circuit model is presented. Finally, the vibration energy harvesting system block diagram, which contains the converter model with its assisted circuit, is explained.

### 4.1. Qualitative Analysis of the Electrostatic MEMS Converter Behavior Based on Its C–x Curve

In this subsection, the electrostatic MEMS converter behavior is qualitatively illustrated based on its *C*–*x* curve. Figure 6a,b demonstrate two fingers of the converter with the input displacement (*x*) caused by the input vibration signal and the converter *C*–*x* curve, respectively.

To explain the converter behavior based on its *C*–*x* curve, from Figure 6a,b, assume that the converter starts moving from its rest position at *d_nom._* (*C_min_*). When the input displacement (*x*) increases in the positive *x*-direction at which the converter fingers move to the right, *d_nom._* decreases to *d_min_*, and the converter capacitance increases till it reaches *C_max_*. Such an operation requires a quarter cycle of the input displacement. Then, when the input displacement decreases to complete the positive half cycle, the fingers move from *d_min_* to *d_nom._*. Thus, the converter capacitance decreases from *C_max_* to *C_min_*, which also requires another quarter cycle of the input displacement. This operation is represented by branch *I* in Figure 6b. Thus, branch *I* gives a half cycle of the input displacement. From Figure 6b, branch *II* of the converter *C*–*x* curve is presented by moving the fingers from the rest position, *d_nom._* and *C_min_*, in the negative *x*-direction to the left. It moves from *C_min_* to *C_max_* and then returns from *C_max_* to *C_min_*. It is clear that the MEMS converter operation is repeated twice during one complete cycle of the input vibration signal. Further, Figure 6c demonstrates the converter capacitance and voltage variations with respect to the input displacement caused by the input vibration signal. As mentioned in the previous section that the charge constrained technique is used in this work; the converter output voltage is inversely proportional to its capacitance variation, as displayed in Figure 6c.

### 4.2. Behavioral Circuit Model of the Enhanced Electrostatic MEMS Converter

In this subsection, based on the qualitative analysis of the converter behavior presented in the previous subsection, the converter behavioral circuit model is illustrated. In this model, a sampling of the converter *C*–*x* curve using both its *C_max_* and *C_min_* is performed. Figure 7 demonstrates the converter behavioral circuit model. The model is built up based on two types of circuit elements. The first type is the circuit elements which are based on the converter design, *C_max_* and *C_min_*. The second type is the elements that are necessary for accurately representing the converter behavior with respect to the input vibration signal, *L*_1_, *I*, *S*_1_, and *S*_2_. The values of *C_max_* and *C_min_* with the deposition of Ta_2_O_5_ are used, namely 0.22 µF and 15.4 nF. The coil, *L*_1_, is used as a charge pump technique. It is the model circuit element that is responsible for the charge transfer operation from *C_max_* to *C_min_*. The designed value of *L*_1_ will be discussed in the coming section. Switches *S*_1_ and *S*_2_ are used to complete the charge transfer operation from *C_max_* to *C_min_* through *L*_1_. The current source, *I*, is used to represent the work undertaken by the converter to convert the input vibration signal into electricity.

Figure 8 [38,39,40] demonstrates the block diagram of the vibration energy harvesting system which is required for reinvestigating the proposed model. This diagram will be utilized for the validation and evaluation of the converter behavioral circuit model. The converter model with its assisted circuit, the power conditioning circuit, is illustrated. This circuit is important for completing the converter operation [33].

Thus, the validation and evaluation of the converter behavioral circuit model will be accomplished through the model interaction with its accompanied circuit. Moreover, the required pulses which control the converter operation will be accurately determined.

As shown in Figure 8, a reservoir capacitor, *C_res_*, must be used to give the initial charge for the electrostatic MEMS converter at the beginning of each input vibration cycle. Thus, it is considered the system supply. Moreover, when the converter exerts work to convert the input mechanical vibration signal into electricity, *C_res_* stores the converted energy at the end of each input vibration cycle. So, it acts as a storage element. The value of *C_res_* must be greater than the converter’s maximum capacitance to fulfill its required operation as the system supply and storage element [33]. In this work, the value of *C_res_* was assumed to be 22 μF which was sufficient for achieving its required operation, as illustrated in the coming section.

Based on the normal operation of any capacitor, if the converter charges directly from *C_res_*_,_ it will charge by only half of its stored energy. Thus, half of the converted output power will be lost [52]. The coil, *L*, which is shown in Figure 8, is used to overcome this issue. It acts concurrently as a charge storage and transfer element. At the beginning of the input vibration cycle, *L* charges from *C_res_*. So, it stores the charge till it reaches the required value to be transferred to the converter as an initial charge without any leakage. In our simulation, *L* was assumed to be 45 μH. This value was required for achieving the resonant frequency with *C_max_* at the vibration frequency. Thus, the inductor can discharge in a quarter cycle of the input vibration. This means that the efficient charge transfer operation through the system could be achieved, as explained in the coming section.

The power switches *S_I_* and *S_II_* are essential for completing the charge transfer operation through the overall system. In this work, the controlling pulses of *S_I_* and *S_II_* operations are generated from a pulse generator. The specifications of such pulses, which are responsible for controlling the whole charge transfer operation through the system, are accurately illustrated and determined in the coming section. The design of the converter controlling circuit based on such specifications can be easily achieved.

## 5. Validation and Evaluation of the Electrostatic MEMS Converter Behavioral Circuit Model

In this section, the converter behavioral circuit model will be investigated through the interaction of the model with the power conditioning circuit explained in the previous section. Firstly, the converter model is qualitatively validated by illustrating the charge transfer operation between system components. Secondly, the proposed model is quantitatively evaluated by simulating its interaction with the power condition circuit. Finally, the required controlling pulses for the converter operation are accurately specified.

### 5.1. Qualitative Validation of the Converter Model

Figure 9a demonstrates the converter behavioral circuit model with its power conditioning circuit. As explained in the previous section, the controlling pulses are assumed to be generated from a pulse generator. Thus, the specifications of the required controlling pulses for the converter operation will be determined based on the model interaction with the power conditioning circuit. Some switches must be added to Figure 9a to complete the interaction between the converter model and the power conditioning circuit based on the controlling pules. Figure 9b demonstrates the final circuit with all of the necessary added circuit elements used for validating and evaluating the converter model [38,39,40].

To be able to validate the converter model qualitatively, the required transitions between the model and *C_res_*, which completely represent the vibration energy harvesting system operation, must be illustrated. Such transitions are based on the required controlling pulses for each switch included in Figure 9b. Figure 10 demonstrates the required stages for completing the operation of the vibration energy harvesting system.

There are three stages required for completing such an operation. The first stage is the charging of the converter from the system supply, *C_res_*, by *V_min_* when its capacitance is at *C_max_* at the beginning of the input vibration cycle. Referring to Figure 9b, such a stage is carried out by charging the converter with *V_min_* at *C_max_* from *C_res_* through *L*. *S_I_*, *S_II_*, and *S*_3_ are responsible for such operation. Figure 10a demonstrates the controlling pulses of the switches, *S_I_*, *S_II_*, and *S*_3_, along with the charge transfer operations required for the first stage. It is obvious that *L* is charging by the converter initial charge from *C_res_* when *SI* is on, which is controlled by Pulse 1, *P*_1_. The amount of the converter required initial charge is determined based on the pulse width of *P*_1_, *PW*_1_. Then, *L* discharges in *C_max_* and *C_max_* charges to *V_min_* from *L* when *S_II_* and *S*_3_ are on, which is controlled by pulse 2, *P*_2_.

The second stage is the conversion of the input vibration signal into electricity using the converter model. Recalling the converter *C*–*x* curve in Figure 6b, firstly, the converter capacitance must decrease from *C_max_* to *C_min_*. Thus, its voltage must increase from *V_min_* to *V_max_*. The current source *I* is added to complete the charging of *C_min_* to *V_max_*. Thus, it is used to represent the converter exerted work through the conversion operation. *S*_1_ and *S*_2_ are responsible for such operation, as will be explained in detail in the coming section. Figure 10b demonstrates the switches controlling pulses along with the charge transfer operations required for the second stage. It is clear that, during pulse 3, *P*_3_, *C_max_* discharges in *L*_1_ through *S*_1_. Then *L*_1_ charges *C_min_* to *V_Cmin_* during pulse 4, *P*_4_, through *S*_2_. Finally, the current source, *I*, complete the charging of *C_min_* till *V_max_*.

The third and final stage is the transfer of the gained energy caused by the conversion operation from the converter to the system storage element. Such an operation is carried out using *L*, *S*_4_, *S*_5_, and *S*_6_. Figure 10c demonstrates such a stage. As shown in the figure, *L* charges from *C_min_* through pulse 5, *P*_5_, when *S*_4_ and *S*_5_ are on. Then *C_res_* charges from *L* by the gained energy from the conversion operation through pulse 6, *P*_6_ when *S*_6_ is on. The diode, D, is added as system protection and to complete the transfer operation.

### 5.2. The Quantitative Evaluation of the Converter Model

In this subsection, the converter model operation is quantitatively evaluated using the OrCAD simulator. The essential three stages required to accurately represent the vibration energy harvesting system operation with all the required details are illustrated. The detailed operation with the accurate specification of each controlling pulse required for controlling switches operation during each transition is illustrated. Finally, the specifications of the converter operation controlling pulses are precisely determined. Moreover, the voltage of *C_res_* at the beginning and the end of the input vibration cycle is simulated and illustrated.

#### 5.2.1. Charging the MEMS Converter with the Initial Charge

In this subsection, the first stage of the vibration energy harvesting operation is illustrated. Figure 11 represents the simulation results for such stage by using OrCAD 16.6; assuming that the electrostatic MEMS converter capacitance is at *C_max_*. At the beginning of the input vibration cycle, the converter must be charged with the initial charge to start the conversion operation. *C_res_*, which acts as the system supply, is responsible for charging the converter with the required initial charge. Thus, the converter needs to be charged with a minimum voltage, *V_min_*, from *C_res_*. Referring to Figure 10, as mentioned in the previous sections, *L* is used as a charge storage element to save the losses which occur from the direct charge of the converter from *C_res_*. Thus, it stores from *C_res_* the required energy for charging the converter with *V_min_*. This charge transfer operation between *L* and *C_res_* is controlled by *S_I_*. The specification of *S_I_* controlling pulse is responsible for limiting the charge transfer from *C_res_* to *L* with the value of *V_min_*.

To calculate *V_min_*, based on the charge constrained conversion operation [33], the charge (*Q*) will be constant during the transfer operation. Thus, *V_min_* is determined by referring to Equation (5). Knowing that *C_max_* = 0.22 µF, *C_min_* = 15.4 nF, and *V_max_* = 8 V, then *V_min_* is 0.56 V. Further, to determine the specifications of the controlling pulse *P*_1_, which controls S_I_ operation, the pulse width of *P*_1_ must be calculated from the maximum current required to charge *C_max_* to *V_min_*. It means that when the charge is transferred from *L* to *C_max_*, the kinetic energy on *L* is changed to potential energy on the capacitor, *C_max_*. As both *C_max_* potential energy and *L* kinetic energy must be equal, as indicated in Equation (9),
(9)$$½C_{max}V_{min}^{2}=½LI_{Lmax}^{2}$$

Given the following values *V_min_* = 0.56 V, *C_max_* = 0.22 µF, *L* = 45 µH, *I_Lmax_* is found to be 39 mA.

To determine the pulse width of *P*_1_, which controls the operation of *S_I_*, *PW*_1_, *L* must be charged from *C_res_* by a maximum current. At the end of the charging cycle, the coil voltage, *V**_L_*, must reach *V_supply_*, which is the voltage of *C_res_*. Thus, *PW*_1_ is calculated from Equation (10),
(10)$$V_{L}=(LI_{Lmax})/PW_{1}$$

In this simulation, *V_Cres_* was assumed to be 3.3 V which is the supply limiting voltage of the 0.35 µm technology used for the power switches [47]. Thus, *PW*_1_ was calculated to be 0.53 µs. To let *L* completely charge to the required maximum current, *PW*_1_ was set to be 0.537 µs. Figure 11a illustrates the simulation result of charging *L* from *C_res_* by the initial charge required to be placed on the converter at the beginning of the input vibration cycle.

It is clear that the simulated value of *I_Lmax_* is 39.027 mA which agrees with the calculated value. In addition, *V_Cres_* decreases to 3.2999 V, which clarifies that *C_res_* acts as a supply for the system. Then, the stored energy on *L* is transferred to charge *C_max_* by *V_min_* through *S_II_*. As *L* must be completely discharged in *C_max_*, *V_L_* must reach zero volts. This operation requires a quarter of the resonant period of the *L_Cmax_* resonant circuit. Thus, the required pulse width of *P*_2_, *PW*_2_, which is required to control the operation of *S_II_* and *S*_3_, is calculated from Equation (11)
(11)$$PW_{2}=\frac{T_{res}}{4}$$
where, *T_res_* is the resonance time constant. So, the resonance frequency (*f_res_*) could be calculated from
(12)$$f_{res}=\frac{1}{T_{res}}=\frac{1}{2\pi \sqrt{LC_{max}}}$$

For *L* = 45 µH and *C_max_* = 0.22 µF, *f_res_* = 50 MHz and thus, *PW*_2_ = 5 μs. To avoid the over or under discharging of *C_max_* from the coil, a slight delay must exist between *P*_1_ and *P*_2_. Thus, *PW*_2_ is assumed to be 0.538 μs. Figure 11b illustrates the simulation result of *S_II_* and *S*_3_ controlling pulse, *P*_2_, and the voltage on *C_max_* and the current *I_L_*. From the figure, it is obvious that the voltage on *C_max_* is 0.552 V which agrees with the calculated value. Moreover, *I_L_* decreases to 0 A at the end of *PW*_2_. It means that *L* transfers all the required initial charges to *C_max_*.

#### 5.2.2. Vibration Energy Conversion Representation by the Converter Model

Figure 12 shows the second stage of the vibration energy harvesting system operation. The conversion of the vibration energy by using the converter model is illustrated. To complete the conversion operation, recalling Figure 6b, the converter capacitance must change from *C_max_*, which has *V_min_*, to *C_min_*, which has *V_max_*. Then the converter capacitance must change again from *C_min_* to *C_max_* to complete the positive half cycle of the input vibration. Such a process is carried out by the converter model as follows. Firstly, the initial charge on *C_max_* must be transferred to *C_min_* through *L*_1_. Thus, *L*_1_ must charge to the maximum current, *I_L_*_1*max*_, from *C_max_* through *S*_1_.

By charging *L*_1_ from *C_max_*, the coil potential energy is transferred to kinetic energy, which is expressed by Equation (13)
(13)$$½C_{max}V_{min}^{2}=½L_{1}I_{L1max}^{2}$$

Using the values: *C_max_* = 0.22 µF, *V_min_* = 0.56 V, and *L*_1_ = 45 µH. *I_L_*_1*max*_ is found to be 39 mA.

Theoretically, the pulse width of *P*_3_, *PW*_3_, which controls the operation of *S*_1_, must equal the pulse width of *P*_2_ as *L*_1_ equals *L*. Slight delay must be added between *P*_2_ and *P*_3_ to avoid the overlap between the operations. Thus, *PW*_3_ = 5 µs. Figure 12a shows the charging of *L*_1_ during *PW*_3_. It is clear that *I_L_*_1*max*_ is 39 mA, which agrees with the calculated value. Then, *C_min_* must be charged by *V_Cmin_* by transferring the charge on *L*_1_ to *C_min_* through *S*_2_. Thus, the kinetic energy of *L*_1_ is transformed to potential energy on *C_min_* as expressed in Equation (14)
(14)$$½C_{min}V^{2}{}_{Cmin}=½L_{1}I_{L1max}^{2}$$

Substituting *C_min_* = 15.4 nF, *L*_1_ = 45 µH, and *I_L_*_1*max*_ = 39 mA in Equation (14), *V_Cmin_* was found to be 2.1 V.

Next, the pulse with of *P*_4_, *PW*_4_, required to control the operation of *S*_2_, was calculated from Equation (11) to be *PW*_4_ = (¼) *T_res_* = 1.3 μs given *L*_1_ = 45 µH and *C_min_* = 15.4 nF. The resonance frequency was calculated based on the following equation,
(15)$$f_{res}=\frac{1}{T_{res}}=\frac{1}{2\pi \sqrt{L_{1}C_{min}}}$$

In this step, *L*_1_ must be directly discharged in *C_min_*; thus, there is no need for inserting a delay between *P*_3_ and *P*_4_. If any delay exists, there will be no path for *L*_1_ to discharge in *C_min_*. Figure 12b shows the charging of *C_min_* from *L*_1_ during *PW*_4_ through *S*_2_.

It is clear that *V_Cmin_* was 2.1 V which agreed with the calculated value. To enable the converter to complete the conversion operation, *V_Cmin_* must reach *V_max_*, which was 8 V in this work. Thus, recalling Figure 7, a current source (*I*) was added to the converter model. This added source was responsible for representing the work undertaken by the converter, which was required to boost *V_Cmin_* to *V_max_* when the converter capacitances changed from *C_max_* to *C_min_*. Both magnitude and pulse width of the current source, I, were calculated based on the extra charge, *Q_extra_*, required to boost *V_Cmin_* to *V_max_*. *Q_extra_* was calculated from Equation (16)
(16)$$Q_{extra}=\Delta V_{C}{}_{min}C_{min}=(V_{Cmin(final)}-V_{Cmin(initial)})C_{min}$$

Substituting in Equation (16) by: *V_Cmin (initial)_* = 2.1 V, *V_Cmin (final)_* = *V_max_* = 8 V and *C_min_* = 15.4 nF, thus *Q_extra_* is calculated to be 90.86 nC. As *Q_extra_* = *I_max_t*, where *I_max_* is the peak of the current source, and *t* is its pulse width. The peak current *I_max_* was assumed to have the maximum current of *L* and *L*_1_, which was 39 mA. From Figure 12c, it is apparent that the voltage on *C_min_* increases from *V_Cmin_*, 2.1 V, to *V_max_*, because of the current source, as explained before.

#### 5.2.3. Transfer the Gained Energy from Conversion Operation back to *C_res_*

In this subsection, the final stage required to completely represent the vibration energy conversion operation is illustrated. At the end of the input vibration cycle, the gained energy from the conversion operation is transferred to the system storage element, *C_res_*, through *L*. Firstly, the gained energy was transferred to *L* from *C_min_* through *PW*_5_ when *S*_4_ and *S*_5_ were ON. The discharging of *C_min_* in *L* follows the same concept of discharging *L*_1_ in *C_min_*. Thus, the required pulse width of *P*_5_, *PW*_5_, for discharging *C_min_* in *L* equals the pulse width of *P*_4_, *PW*_4_, which was 1.3 μs. By charging *L* from *C_min_*, the potential energy was transferred to kinetic energy. So, following the same approach as before, one can find *I_L_* to be 147.99 mA.

Figure 13a shows the discharging of *C_min_* in *L* during *PW*_5_ through *S*_4_ and *S*_5_ and the coil current, which agrees with the calculated value, 148 mA. Secondly, the gained energy which is stored in *L* is transferred to the system storage element, *C_res_*. Thus, *L* discharges in *C_res_* through *P*_6_ when *S*_6_ is on. Figure 13b shows the transferring of the gained energy from *L* to *C_res_*. It is obvious that *V_Cres_* increases from 3.299 V to 3.305 V.

It is important to calculate the gained energy caused by the conversion of the input vibration signal into an electric signal using the converter model. In order to calculate the gained energy by *C_res_* from the conversion process, the initial voltage on *C_res (Vinitial)_* has to be specified. This is the voltage on *C_res_* at the start of the input vibration cycle. In addition, the final voltage on *C_res (Vfinal)_* has to be specified. This is the voltage on *C_res_* at the end of the input vibration cycle. As the input vibration frequency in this study is 2.5 kHz, thus the input vibration cycle is 0.4 ms. From Figure 13b, it is clear that the values of the *V_initial_* and *V_final_* are 3.299 V and 3.305 V, respectively. The gained energy by *C_res_* is given by Equation (17)
(17)$$E_{gained}=E_{final}-E_{initial}=½\;C_{res}\Delta V_{Cres}^{2}$$

Where $\Delta V_{Cres}^{2}$ is given by, $\Delta V_{Cres}^{2}=V_{final}^{2}-V_{initial}^{2}$, taking the previously listed value of *C_res_*, *E_gained_* is found to be 0.44 µJ. Moreover, the gained power by *C_res_* from the system is given by Equation (18)
(18)$$P_{gained}=2fE_{gained}$$

Given *f* = 2.5 kHz, *E_gained_* = 0.44 µJ, *P_gained_* was found to be 2.2 mW which gives a good agreement with the calculated value of *P_out_* from the converter, which was 2.3 mW.

Finally, Figure 14 illustrates the simulation of *V_Cres_* for multiple cycles of the input vibration signal. Such a figure emerges as the major contribution of *C_res_* to the vibration energy harvesting system. It verifies that *C_res_* acts as the system supply at the beginning of the input vibration cycle. Therefore, *V_Cres_* decreases by the value required initially to charge the converter. At the end of the input vibration cycle, *C_res_* acts as the system storage element. So, *V_Cres_* increases by the value of the gained energy from the conversion operation.

The proposed behavioral circuit model is applied to a traditional simulated electrostatic in-plane gap closing MEMS converter from reference [20] and to another practically fabricated converter from reference [26] to provide a calibration study and to validate the accuracy of our presented model. The interaction between the model of the traditional converters and the system is simulated. Figure 15a,b show the simulation results of *V_cres_*, which is the main measurement of the system’s efficient performance during one cycle of the input vibration frequency for both cases.

From the simulation results, based on the value of the initial and final voltage on *C_res_* and by using Equations (17) and (18), the gained energy and the output power from the system were calculated for both calibrated cases. Table 2 lists the main input parameters used in our simulation that were extracted from the published studies [20,26], while Table 3 presents a comparison between our used converter case study and both the simulated and fabricated converters. The relative absolute percentage error is also shown, indicating errors that are less than 10%.

## 6. Conclusions

To overcome the difficulty of fabricating the electrostatic MEMS converter due to the unavailability of fabrication facilities, we have investigated a modified behavioral circuit model for the MEMS converter based on its *C*–*x* curve. Such a model has the advantage of being adaptive; thus, it can be applied to any MEMS converter system. Additionally, the model enables the implementation and testing of the MEMS converter along with the overall vibration energy harvesting system using commercially available off-shelf components. Thus, the designed converter performance can be cheaply tested and evaluated before going through the costly fabrication procedures.

Moreover, a proposed technique is presented for which the conventional in-plan gap-closing electrostatic MEMS converter performance is enhanced by depositing Ta_2_O_5_ on its sidewall fingers. The main equations which govern the converter performance are illustrated. One of the main basic key parameters of the vibration energy harvesting system is the maximum voltage *V_max_*, which is thoroughly investigated. Based on the simulation results using the MATLAB PDE tool, the optimum *V_max_* is found to be 8 V. Thus, the output power of the enhanced converter is calculated to be 2.3 mW which is considered an appreciable enhancement in comparison with recently cited work.

Next, the behavioral model is applied to the proposed enhanced structure. Firstly, the model is qualitatively validated by illustrating its behavior during the three essential stages required for successfully achieving the vibration energy conversion operation. Then, the model is quantitatively evaluated using the Orcad simulator. Based on the simulation results, the required controlling pulses for the converter operation are accurately specified, which gives the advantage of the ease of implementing the converter controlling circuit. Moreover, the simulation results emphasize that the generated output power from the converter model based on its interaction with the system shows a good agreement with the analytically calculated output power of the converter, 2.3 mW. Thus, the converter behavioral circuit model accurately accomplished the vibration energy conversion operation. Finally, the proposed behavioral circuit model is applied to traditional simulated and fabricated electrostatic in-plane gap closing MEMS converter to provide a calibration study and to validate the accuracy of our presented model; it gives good agreement with such cases; thus, the model accuracy is verified. In our future work, we intend to practically implement and test the proposed converter behavioral circuit model along with its power conditioning circuit. Furthermore, the system performance will be examined using different loads. Moreover, the enhanced converter will be simulated by FEM under the MATLAB environment to optimize its technological, physical, and mechanical parameters to improve its performance.

## Figures and Tables

**Figure 1 micromachines-13-00868-f001:**
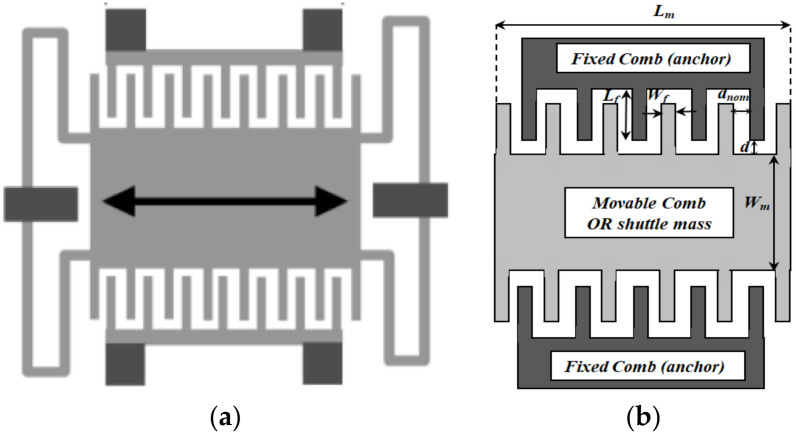
(**a**) The conventional gap-closing electrostatic MEMS converter and (**b**) A detailed description of the converter comb drive parameters [42].

**Figure 2 micromachines-13-00868-f002:**
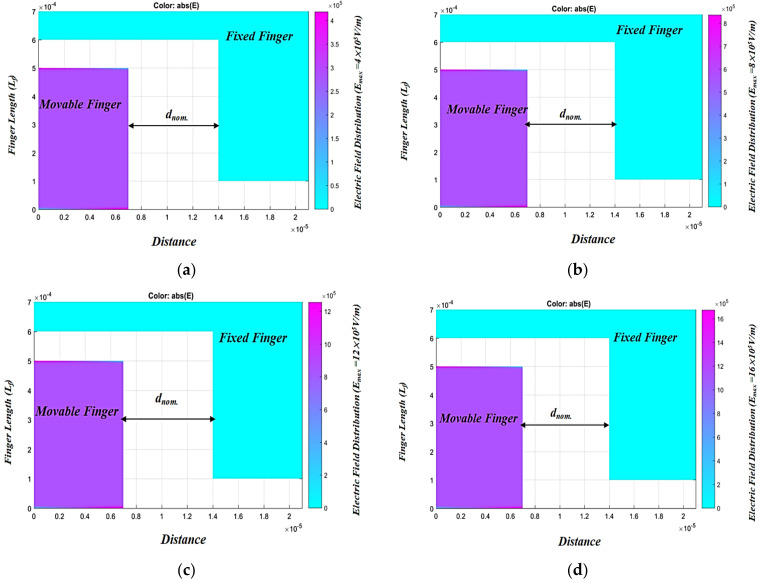
Electric field distribution between the converter fingers at different input voltage, *V_ip_* (**a**) *V_ip_* = 2 V, (**b**) *V_ip_* = 4 V, (**c**) *V_ip_* = 6 V, (**d**) *V_ip_* = 8 V, and (**e**) *V_ip_* = 10 V.

**Figure 3 micromachines-13-00868-f003:**
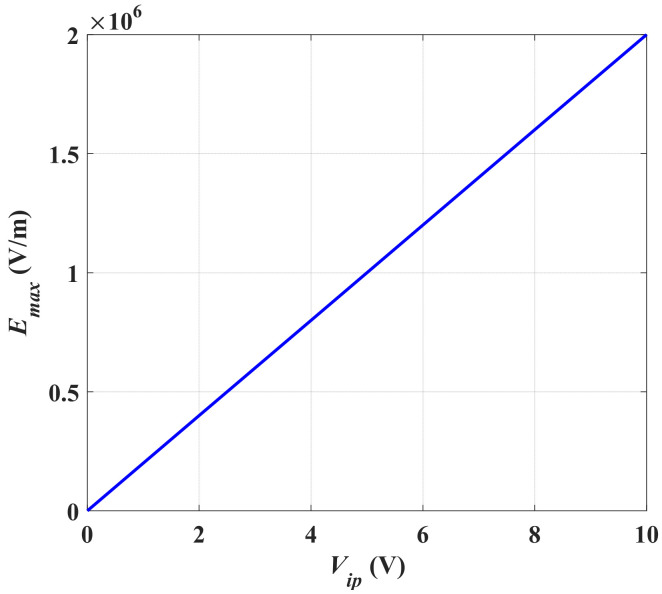
*E_max_* between the converter fingers at different *V_ip_*.

**Figure 4 micromachines-13-00868-f004:**
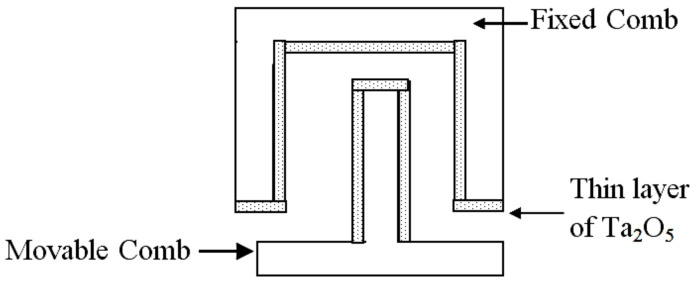
Demonstration of the converter fingers with the deposition of Ta_2_O_5_.

**Figure 5 micromachines-13-00868-f005:**
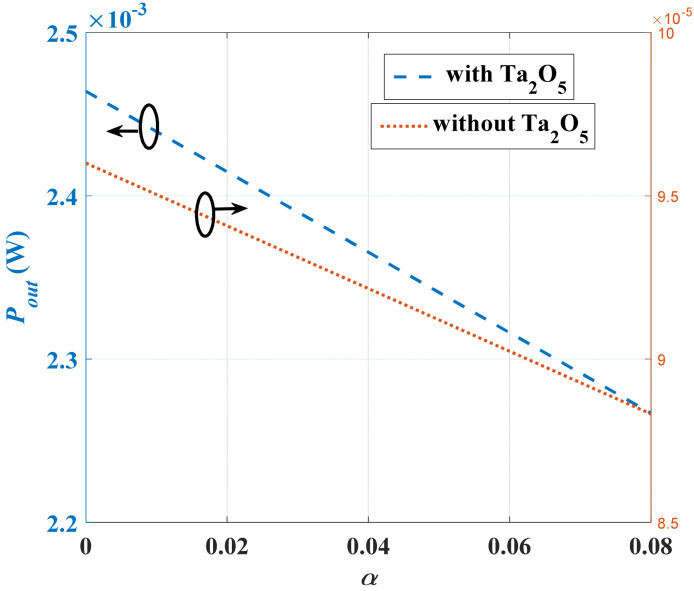
*P_out_* versus *α* at two different cases (with and without the deposition of Ta_2_O_5_).

**Figure 6 micromachines-13-00868-f006:**
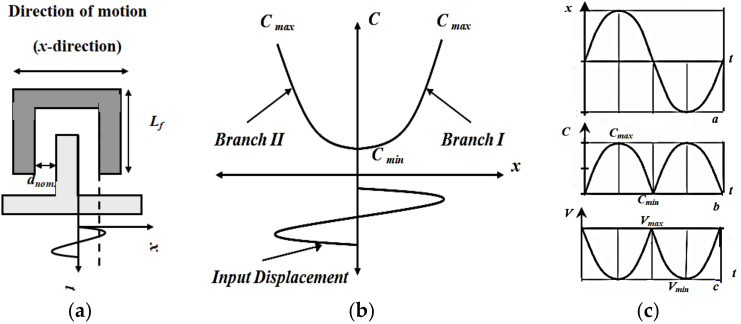
Qualitative analysis of the converter behavior (**a**) The converter two fingers representation, (**b**) The converter *C*–*x* curve, and (**c**) The input displacement (*x*), changes in capacitance and converter output voltage.

**Figure 7 micromachines-13-00868-f007:**
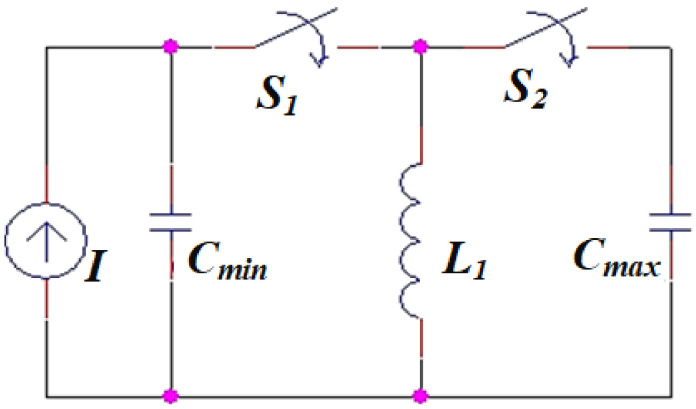
The behavioral circuit model of the electrostatic MEMS converter.

**Figure 8 micromachines-13-00868-f008:**
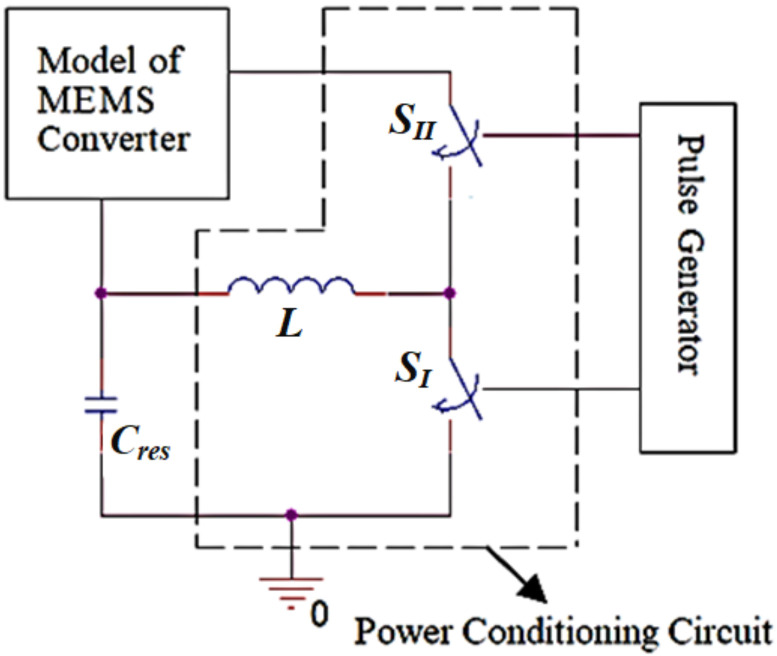
The block diagram for the vibration energy harvesting system.

**Figure 9 micromachines-13-00868-f009:**
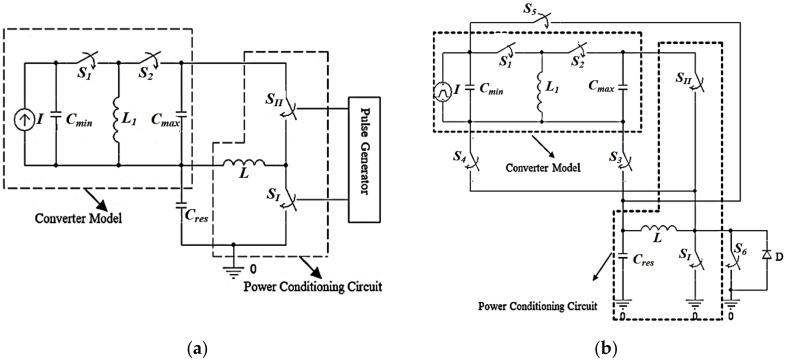
Demonstration of the converter behavioral circuit model with its power conditioning circuit (**a**) Converter behavioral circuit model with its power conditioning circuit, and (**b**) Final circuit with all the necessary added elements.

**Figure 10 micromachines-13-00868-f010:**
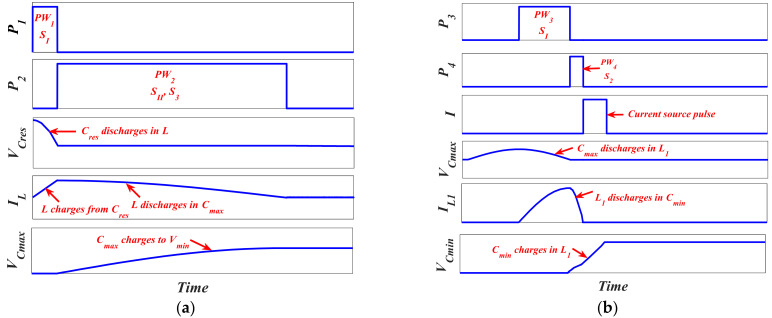
The three stages of the vibration energy harvesting system operation (**a**) The first stage, (**b**) The second stage, and (**c**) The third stage.

**Figure 11 micromachines-13-00868-f011:**
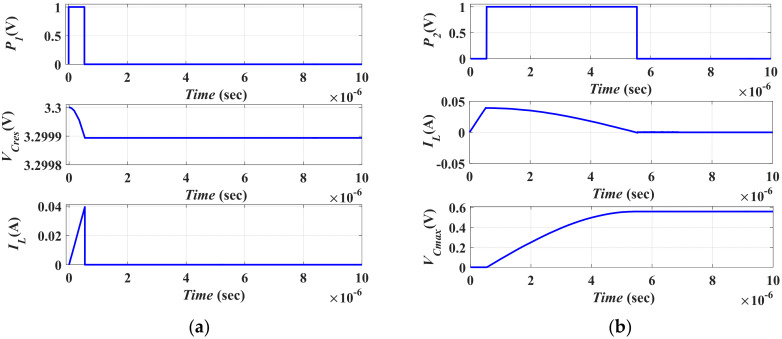
The first stage simulation results (**a**) The charging operation of *L* from *C_res_*, and (**b**) The Simulation result of *P*_2_, and the voltage on *C_max_* and *I_L_*.

**Figure 12 micromachines-13-00868-f012:**
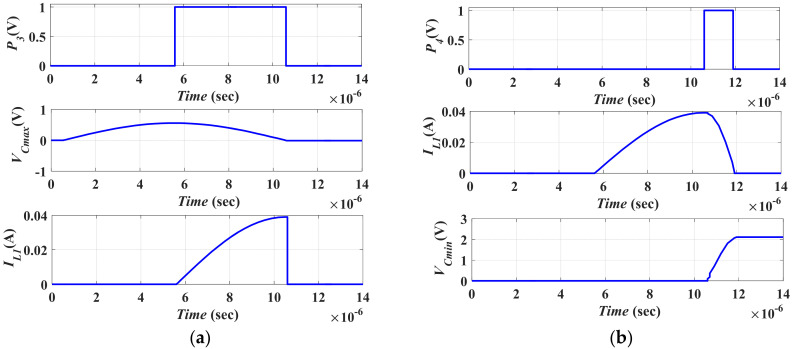
The second stage simulation results (**a**) The Simulation result of *P*_3_, and the charging current of and *L*_1_, (**b**) Charging of *C_min_* from *L*_1_ during *PW*_4_ through *S*_2_, and (**c**) Boosting *V_Cmin_* to *V_max_* by the effect of *I*.

**Figure 13 micromachines-13-00868-f013:**
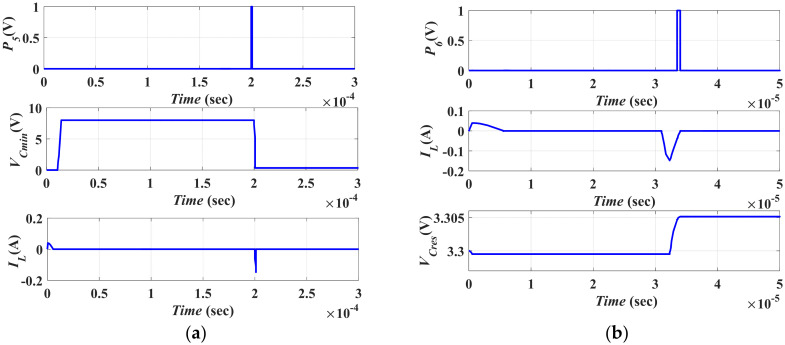
The third stage simulation results (**a**) The discharging of *C_min_* in *L* during *PW*_5_ through *S*_4_ and *S*_5_, and (**b**) Transferring the gained energy from *L* to *C_res_* during *PW*_6_.

**Figure 14 micromachines-13-00868-f014:**
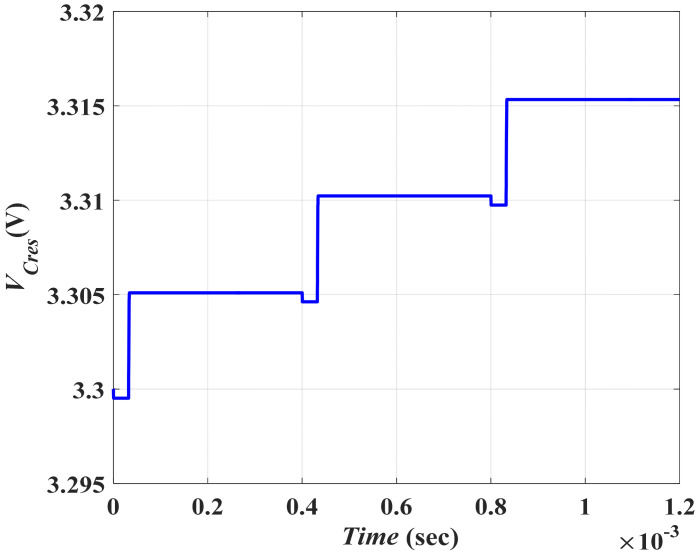
Variation of *V_Cres_* during multiple cycle of the input vibration signal.

**Figure 15 micromachines-13-00868-f015:**
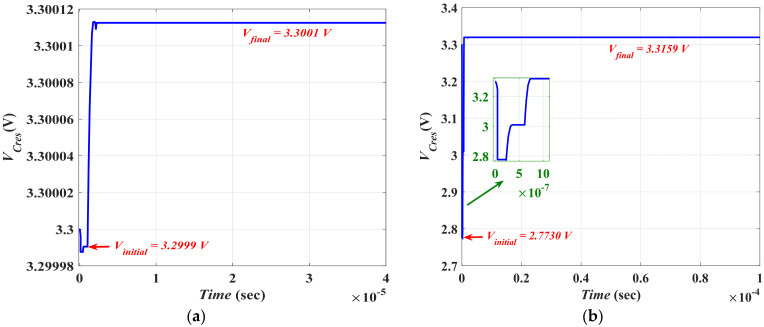
The reservoir capacitor voltage (**a**) for the simulated converter (**b**) for the fabricated converter.

**Table 1 micromachines-13-00868-t001:** Main technological parameters of the conventional gap-closing electrostatic MEMS converter comb drive for the case study.

Symbol	Technological Parameter	Value
*L_f_*	Finger Length	512 µm
*W_f_*	Finger Width	7 µm
*d_nom._*	Nominal Gap between Fingers at rest position	7 µm
*d*	Dielectric Distance	50 µm
*L_m_*	Shuttle Mass Length	1 cm
*W_m_*	Shuttle Mass Width	0.3 cm
*L_terminal_*	Positive Terminal Length	100 µm
*W_terminal_*	Positive Terminal Width	100 µm
*N_g_*	Number of fixed or movable fingers	476
*t*	Device thickness	500 µm

**Table 2 micromachines-13-00868-t002:** Input parameters of different designs of electrostatic MEMS converters. Also, our design input parameters are shown for comparison.

Input Parameters	REF [20]	REF [26]	Our Work
*C_max_*	800 nF	290 pF	0.22 µF
*C_min_*	20 nF	40 pF	15.4 nF
*V_max_*	100 V	10 V	8 V
*f_o_*	120 Hz	250 Hz	2.5 kHz
*C_res_*	200 nF	5 nF	22 µF

**Table 3 micromachines-13-00868-t003:** Comparison between output parameters of different studies and our model calculations showing the relative absolute percentage errors of results with respect to ours.

	*E_gained_*	*P_gained_*	Method
**REF [20]**	0.175 µJ	42 µW	Analytical modeling
**Our Model**	0.189 µJ	45 µW	Circuit simulation
**Δξ (%)**	8.00	7.14	
**REF [26]**	8.8 nJ	4.4 µW	Experimental
**Our Model**	8.2 nJ	4.1 µW	Circuit simulation
**Δξ (%)**	6.82	6.82	

## Data Availability

No new data were created or analyzed in this study. Data sharing is not applicable to this article.

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
