# Peer review of "Validation and Evaluation of a Behavioral Circuit Model of an Enhanced Electrostatic MEMS Converter"

_micromachines, 2022, doi:10.3390/mi13060868_

Round 1

Reviewer 1 Report

After reviewing your article, the overall performance is good. More comparison among piezoelectric, electromagnetic, and electrostatic harvesters must be added.

The grammar and content must be revised more. For instance: Lines163, 174, 437, 489, 561, and 566 are not consistent.

In the technical part, I have some concerns and you must step by step illustrate them.

  1. In Fig. 3, the sensed data must be labeled with shape, not a straight line. The linear regression R2 must be shown.
  2. In Line 239, I didn’t see the information related to Table 2.
  3. In Fig. 5, two curves with and without Ta2O5 in comparison are good, but the real performance with Ta2O5 deposition are not estimated, such as the thickness and leakage. These parameters will deeply impact the energy converting efficiency.
  4. In Fig. 6(a), it’s a simulation plot. Is it repeatable? If yes, how can you estimate the max. and min. deviation range and the acceptable range?
  5. In Fig.8, the inductance is estimated as 45 uH. What is the core reason? If this factor is changed, what is the main impact related to Eqs.(9) to (15)?
  6. In Fig. 14, the explanation for variation of VCres during multiple cycle of the input vibration signal is not clear. The VCres has some sunken characteristics. You must clearly illustrate them and provide some possible solutions to enhance the gained energy.

Author Response

Reviewer 1:

Comments and Suggestions for Authors

After reviewing your article, the overall performance is good.

Thank you so much. Really, the comments and suggestions are very helpful to improve our work. We hope that our reply and discussion cover the points mentioned in this review satisfactorily.

More comparison among piezoelectric, electromagnetic, and electrostatic harvesters must be added.

Our concentration in this work is on electrostatic harvesters. We added some sentences on the piezoelectric and electromagnetic harvesters in our revised manuscript to highlight the difference between these types of harvesters and the reasons beyond our choice of electrostatic harvesters in particular. We added the following paragraph:

“In an electromagnetic energy harvester, the losses caused by its coil resistance are one of its major drawbacks. In addition, it requires complex fabrication processes which have low process compatibility [14]. Electrostatic harvesters, on the other hand, can be easily fabricated using standard micro-machining processes [15–18]”

Also, we updated the reference list by adding the following references which support the comparison:

  1. Fang, S.; Fu, X.; Du, X.; Liao, W. H. A music-box-like extended rotational plucking energy harvester with multiple piezoelectric cantilevers. Applied physics letters 2019, 114(23), 233902, https://doi.org/10.1063/1.5098439.
  2. Jung, I.; Shin, Y. H.; Kim, S.; Choi, J. Y.; Kang, C. Y. Flexible piezoelectric polymer-based energy harvesting system for roadway applications. Applied Energy 2017, 197, 222-229, https://doi.org/10.1016/j.apenergy.2017.04.020.
  3. Tan, Q.; Fan, K.; Tao, K.; Zhao, L.; Cai, M. A two-degree-of-freedom string-driven rotor for efficient energy harvesting from ultra-low frequency excitations. Energy 2020, 196, 117107, https://doi.org/10.1016/j.energy.2020.117107.
  4. Tao, K.; Wu, J.; Tang, L.; Xia, X.; Lye, S. W.; Miao, J.; Hu, X. A novel two-degree-of-freedom MEMS electromagnetic vibration energy harvester. Journal of Micromechanics and Microengineering 2016, 26(3), 035020, ‏ https://doi.org/10.1088/0960-1317/26/3/035020.

The grammar and content must be revised more. For instance: Lines163, 174, 437, 489, 561, and 566 are not consistent.

The manuscript has been reviewed and re-corrected. We have made careful modifications to the original manuscript, and thorough proofreading is done to minimize typographical and grammatical errors. We carefully take into consideration the main lines focused by the reviewer.

In the technical part, I have some concerns and you must step by step illustrate them.

  1. In Fig. 3, the sensed data must be labeled with shape, not a straight line. The linear regression R2must be shown.

The relation, illustrated in Figure 3, is perfectly linear and we did not apply any fitting equations.

  1. In Line 239, I didn’t see the information related to Table 2.

Duly noted, there was no Table 2 in the main manuscript. This is now corrected.

  1. In Fig. 5, two curves with and without Ta2O5in comparison are good, but the real performance with Ta2O5deposition is not estimated, such as the thickness and leakage. These parameters will deeply impact the energy converting efficiency.

The thickness of Ta2O5 is set to be 0.25 µm, which is equal to dmin. This value is already mentioned in the main manuscript in this paragraph. Such a value of Ta2O5 thickness does not cause leakage at the highest operating voltage. In the modified manuscript, we added the following paragraph which covers this point:

“The selected thickness for Ta2O5 which is 0.25 µm is large enough to avoid the leakage problem. The leakage in the dielectric materials occurs when the dielectric layer thickness is in the range of nm [50–52]”.

Also, we update the reference list by adding the following references:

  1. Ezhilvalavan, S.; Tseng, T. Y. Preparation and properties of tantalum pentoxide (Ta2O5) thin films for ultra large scale integrated circuits (ULSIs) application–A review. Materials Science: Materials in Electronics 1999, 10(1), 9-31.
  2. Paskaleva, A., Atanassova, E., & Dimitrova, T. (2000). Leakage currents and conduction mechanisms of Ta2O5 layers on Si obtained by RF sputtering. Vacuum, 58(2-3), 470-477.‏
  3. Aygun, G., & Turan, R. (2008). Electrical and dielectrical properties of tantalum oxide films grown by Nd: YAG laser assisted oxidation. Thin Solid Films, 517(2), 994-999.‏

Concerning the real performance with Ta2O5 deposition, it is evident that the converter capacitances increase the converter output power. The values of the enhanced converter capacitances, with depositing Ta2O5, are used as input parameters for the behavioral circuit model giving promising operation results as explained in the manuscript.

  1. In Fig. 6(a), it’s a simulation plot. Is it repeatable? If yes, how can you estimate the max. and min. deviation range and the acceptable range?

Figure 6(a), (b) and (c) are not simulation plots. These are demonstrative figures which qualitatively illustrate the converter behavior based on its capacitance as a function of the position x assuming the fingers vibrate inside their space. There is no statistical behavior here. Referring to Figure 6(a), the converter fingers move based on the input displacement caused by the input vibration signal. Thus, the converter capacitances change based on its fingers’ movement. As a result, the converter capacitance changes based on the displacement of the input vibration signal, as illustrated in Figure 6(b). This part is explained in detail in the main manuscript. The converter capacitances changes based on its finger motion caused by the input vibration signal which is a deterministic and well-defined operation in the electrostatic energy harvester following the swinging movement of the fingers inside their stators.

  1. In Fig.8, the inductance is estimated as 45 uH. What is the core reason? If this factor is changed, what is the main impact related to Eqs.(9) to (15)?

In the modified manuscript, we added the following paragraph which explains the core reason for selecting the value of L:

L is assumed to be 45 μH. This value is required for achieving the resonant frequency with Cmax at the vibration frequency, Thus, the inductor can discharge in a quarter cycle of the input vibration. Thus, the efficient charge operation through the system will be achieved as explained in the coming section.”

  1. In Fig. 14, the explanation for variation of VCresduring multiple cycle of the input vibration signal is not clear. The VCreshas some sunken characteristics. You must clearly illustrate them and provide some possible solutions to enhance the gained energy.

Figure 14 illustrates the reservoir voltage, VCres, over multiple cycles of the input vibration signal. At the beginning of a cycle, one charges Cmax from the stored charges in the storage capacitor. Therefore, one observes the slight decrease of its voltage after which its voltage will be kept constant until the discharging of the MEMS capacitor Cmin in it again. Cres acts as the system supply at the beginning of the input vibration cycle. Thus, VCres decreases by the value required initially to charge the converter. At the end of the input vibration cycle, Cres acts as the system storage element. So, VCres increases by the value of the gained energy from the conversion operation in addition to the recharged one. Thus, the sunken in VCres will be compensated again.

Reviewer 2 Report

This work proposes a behavioral circuit model of an enhanced electrostatic MEMS converter for better validation and evaluation. This paper should resolve the following issue to be published.

The main contribution of this paper is very confusing. It is not very clear whether the author claims the novelty of this paper as ‘a new electrostatic MEMS converter with depositing the tantalum pentoxide’ or ‘new behavioral circuit model technique’. For the former, the paper should focus on analyzing the new device and compare the performance with the previous works by showing a performance summary and comparison table. Here, the behavior model should be used as a supporting tool to efficiently develop the new device.

For the latter, the author should compare the accuracy of the new behavioral model with a traditional model. They should be applied to different traditional devices rather than a new device. Also, it should clarify what the accuracy of 100% is. Typically, the measurement result from physical hardware is 100% (reference). However, it seems this paper considers a simulation result as a reference. The definition of accuracy should be clearly defined and quantified.

Author Response

Reviewer 2:

Comments and Suggestions for Authors

This work proposes a behavioral circuit model of an enhanced electrostatic MEMS converter for better validation and evaluation. This paper should resolve the following issue to be published.

We would like to thank the reviewer for the comments and suggestions which are very helpful to improve our work. We hope that our reply and discussion cover the points mentioned in this review satisfactorily.

The main contribution of this paper is very confusing. It is not very clear whether the author claims the novelty of this paper as ‘a new electrostatic MEMS converter with depositing the tantalum pentoxide’ or ‘new behavioral circuit model technique’. For the former, the paper should focus on analyzing the new device and compare the performance with the previous works by showing a performance summary and comparison table. Here, the behavior model should be used as a supporting tool to efficiently develop the new device

The focus of the paper is to build up a proposed behavioral circuit model for the MEMS converter which illustrates the converter behavior. Such a model can be applied to any other electrostatic MEMS converter. As well known, for any device or structure, the circuit or analytical model enables the designer from checking the designed structure performance quickly and transparently before going through the simulation of the time-consuming software, in this case, the FEM simulators under MATLAB environment will be used mainly COMSOL Multiphysics. Thus, it becomes more visible to figure out how to optimize the structure’s technological and physical parameters before starting the simulation using such tools.  

In addition, the practical implementation of this model using the commercially available off-shelf components enables the investigation of the overall vibration energy harvesting system without the need for the unavailable, in our case, and costly fabrication of the converter. Even if the fabrication facilities are available, the cheap implementation of the overall system enables better investigation of the converter behavior and performance. Thus, it enables the designer to accurately consider all technological and physical parameters. Thus, in the paper, we focused on the behavioral model and its validation and evaluation through its interaction with the system. The model achieves an efficient illustration of the converter as explained in the manuscript. Moreover, in the modified manuscript, the proposed behavioral circuit model is applied to a traditional simulated electrostatic in-plane gap closing MEMS converter from reference [20] and to another practically fabricated converter from reference [26] to provide a calibration study and to validate the accuracy of our presented model.

Concerning the electrostatic MEMS converter, which is implemented using the model, we take a case study for the converter. We analytically investigated the converter performance using its main governing equations. Based on such a study, we proposed to enhance the converter performance by depositing Ta2O5 which increases its output power from 0.09 mW to 2.3 mW. In our future work, we are working on simulating this enhanced converter using FEM under MATLAB environment to investigate and optimize its performance. In the modified manuscript we added the following paragraph “In addition, the enhanced converter will be simulated FEM under MATLAB environment to optimize its technological, physical and mechanical parameters to improve its performance”.

In our revised manuscript, we rewrote both the abstract and the conclusions to clarify the idea.

For the latter, the author should compare the accuracy of the new behavioral model with a traditional model. They should be applied to different traditional devices rather than a new device. Also, it should clarify what the accuracy of 100% is. Typically, the measurement result from physical hardware is 100% (reference). However, it seems this paper considers a simulation result as a reference. The definition of accuracy should be clearly defined and quantified.

In the manuscript, the model is applied to the enhanced conventional electrostatic MEMS converter. Concerning the accuracy of the model, it is satisfied by the efficient charge transfer operation between the converter model and the system. The main objective of the model is to represent the conversion operation carried out by the converter model. The output power of the converter model gives a good agreement with the analytically calculated output power of the enhanced converter.

Further, based on the valuable comment, we applied our proposed behavioral circuit model to a traditional simulated electrostatic in-plane gap closing MEMS converter from reference [20] and to another practically fabricated converter from reference [26] to provide a calibration study and to validate the accuracy of our presented model. We added the following parts to the revised manuscript:

Firstly, in the abstract, we added the following statement “Finally, the model accuracy is validated by calibrating its performance with traditionally simulated and fabricated electrostatic MEMS converter”.

Secondly, at the end of the paper, we added the following part:

“The proposed behavioral circuit model is applied to a traditional simulated electrostatic in-plane gap closing MEMS converter from reference [20] and to another practically fabricated converter from reference [26] to provide a calibration study and to validate the accuracy of our presented model. The interaction between the model of the traditional converters and the system is simulated. Figure 15(a) and (b) show the simulation results of Vcres; which is the main measurement of the system’s efficient performance, during one cycle of the input vibration frequency for both cases.

From the simulation results, based on the value of the initial and final voltage on Cres and by using equation 17, the gained energy and the output power from the system are calculated for both calibrated cases. Table 2 lists the main input parameters used in our simulation that are extracted from the published studies [20, 26], while Table 3 presents a comparison between our used converter case study and both the simulated and fabricated converters. The relative absolute percentage error is also shown indicating errors that are less than 10%. 

Round 2

Reviewer 1 Report

Your revised article is better than the initial version. In Eqs.(9), (13), (14), and (17), you must check them again. In Figs. 12, 11, 12, and 15, the figure quality must be improved. The curves can be colored to enhance the research performance.

Author Response

Comments and Suggestions for Authors

Your revised article is better than the initial version.

Thank you so much. The comments and suggestions through the review process were really very helpful to improve our work.

In Eqs.(9), (13), (14), and (17), you must check them again.

Done. The mentioned equations are revised and checked.

In Figs. 12, 11, 12, and 15, the figure quality must be improved. The curves can be colored to enhance the research performance.

In the revised manuscript, we updated all figures to be clearer.

Reviewer 2 Report

Table 3 compares this work with the previous works. It shows the relative absolute percentage errors. However, a reader cannot find the advantage of the proposed technique in this table. Please include a parameter that shows the advantage. For example, smaller model size, reduced simulation time, etc. 

Author Response

Comments and Suggestions for Authors

Table 3 compares this work with the previous works. It shows the relative absolute percentage errors. However, a reader cannot find the advantage of the proposed technique in this table. Please include a parameter that shows the advantage. For example, smaller model size, reduced simulation time, etc. 

The main objective of our proposed model is to be able to check the performance of any electrostatic MEMS converter and have quick and transparent estimation for the direction of optimizing its technological and physical parameters. The models we used to be applied to our model are experimental and analytical models. So, the two models have no simulation. In the revised manuscript, we updated Table 3 to declare this point.